# Improving the Accuracy of Fault Frequency by Means of Local Mean Decomposition and Ratio Correction Method for Rolling Bearing Failure

**Yongqiang Duan [1], Chengdong Wang [1,2,]\*[ID], Yong Chen [1,2][ID] and Peisen Liu [3]**

[1]  School of Automation Engineering, University of Electronic Science and Technology of China, Chengdu 611731, China; 15008257533@163.com (Y.D.); ychencd@uestc.edu.cn (Y.C.)

[2]  Institute of Electric Vehicle Driving System and Safety Technology, University of Electronic Science and Technology of China, Chengdu 611731, China

[3]  School of Mechanical Engineering, Chengdu Technological University, Chengdu 611730, China; liupeisen103@163.com

\*  Correspondence: wangchengdong@uestc.edu.cn

**Abstract:** The fault frequencies are as they are and cannot be improved. One can only improve its estimation quality. This paper proposes a fault diagnosis method by combining local mean decomposition (LMD) and the ratio correction method to process the short-time signals. Firstly, the vibration signal of rolling bearing is decomposed into a series of product functions (PFs) by LMD. The PF, which contains the richest fault information, is selected to perform envelope spectrum analysis by the Hilbert transform (HT). Secondly, the Hilbert envelope spectrum of the selected PF is corrected with the ratio correction method. Finally, higher precision fault frequencies are extracted from the corrected Hilbert envelope spectrum, and then the fault location is accurately determined. The proposed method of this paper can be used in online real-time monitoring technology of rolling bearing failure.

**Keywords:** local mean decomposition; spectrum correction; ratio correction method; frequency accuracy

## 1. Introduction

Rolling bearing is one kind of core parts in rotating machines, which plays an important role in industrial production. Condition monitoring and fault diagnosis of rolling bearing has become an attractive research topic. The purpose of fault diagnosis for rolling bearing is to determine the type of faults, the degree of damage, and the cause of faults. When a fault occurs, rotating machines should stop to repair or replace the faulty rolling bearing in time to avoid serious results. The process of fault diagnosis can be divided into three steps, which are signal acquisition, feature extraction, and diagnosis decision. Among the three steps, feature extraction is the most critical one. The vibration signal of rolling bearing in the fault state usually consists of three parts. The first part is the fault information of rolling bearing with the characteristics of non-stationary, nonlinear, and modulation. The second part is the vibration information of the rotating machines except the faulty rolling bearing. The third part is the noise and interference. Many signal processing methods are used to process the vibration signal, such as demodulated resonance technique (DRT), short time Fourier transform (STFT), wavelet transform (WT), and Hilbert-Huang transform (HHT). While these methods are effective and useful in the fault diagnosis of rolling bearing, there are still some limits. For example, the DRT is difficult to find out the best main resonance frequency band accurately, and the time-frequency window size of STFT is fixed [1]. While WT has a variable time-frequency window, the results are fixed-band signals when the

wavelet basis and decomposition scale are selected [2]. HHT includes empirical mode decomposition (EMD) and HT, and it is a self-adaptive time-frequency analysis method. However, there are some defects in EMD, such as over-envelope, under-envelope, mode mixing, end effect, IMF criterion [3–6], and may produce unexplained negative frequencies after calculating the instantaneous frequency [7]. The application of these methods in the fault diagnosis of rolling bearing suffers from their defects and requirements for data.

In addition to the traditional methods, new techniques and methods are also applied to the fault diagnosis of rolling bearing. With the rapid development of computer technology and machine learning, deep learning algorithms are increasingly applied to the fault diagnosis of rolling bearing. For instance, based on a convolutional neural network (CNN, Address) and a long-short-term memory (LSTM) recurrent neural network, an improved bearing fault diagnosis method is proposed [8]. This method has the advantages of much higher prediction accuracy, faster iteration and more efficient to prevent over-fitting. Besides, the input of this method is the raw sampling signal without any pre-processing or traditional feature extraction. The results showed that the average accuracy rate in the testing dataset reached more than 99%. However, one of its obvious shortcomings is that it requires large amount of computation. A CNN model can learn features from frequency data directly and detect faults of gearboxes [9]. The results indicate that this method is able to learn features adaptive from frequency data and achieve higher diagnosis accuracy. This method may be applied to the fault diagnosis of rolling bearing. A hybrid unsupervised feature selection (HFS) approach demonstrated its effectiveness in the fault diagnosis of rolling bearing [10]. The deep learning models reduce the incompleteness caused by artificial design through self-learning and building feature models. However, in the case of a limited amount of data, the deep learning algorithms cannot make unbiased estimates of the laws of the data. A large amount of data will lead to a long running time of algorithms. In order to ensure the real-time performance, the deep learning models require more optimized algorithms, better hardware and enough data.

In 2005, the British scholar Jonathan S. Smith proposed the LMD on the basis of EMD to deal with non-stationary signals in a self-adaptive way [7]. LMD uses an iterative approach to decompose a signal into a set of product functions (PFs). Each of the PF is the product of an envelope signal and a pure frequency-modulation signal. LMD has a capacity of time-frequency analysis and demodulation analysis for non-stationary signals. Compared with EMD, LMD solved the problem of over-envelope and under-envelope and suppresses the end effect to a certain extent. Scholars have already proved the superiority in LMD [11] and improved the algorithm. LMD has been applied widely to the fault diagnosis of rolling bearing.

Through spectrum analysis, various frequency components contained in the vibration signal can be clearly viewed. However, when performing Fourier transform of signals on computers, which can only process discrete data, it is unavoidable to suffer spectrum leakage and barrier effect [12,13] because of time domain truncation and limited samples. This will lead to errors in frequency, amplitude, and phase, and eventually, affect the extraction accuracy [14]. The frequency resolution $\Delta w$ of a signal has relations to the sampling number $N$ and the sampling frequency $F_s$, based on the formula $\Delta w = F_s/N$. Therefore, the longer the signal length is, the better the frequency resolution can be reached. However, this leads to an increase in the cost and time of computer calculation. It is always needed to analyze signals as fast as possible and reduce the calculating time under the premise of ensuring calculation accuracy, especially in online real-time monitoring application. Therefore, it is considered to correct the spectrum based on less sampling number of raw data. This method can make the spectrum close to the true value to the maximum extent in the case of ensuring the extraction accuracy of fault frequency.

Accurate frequency estimation is one of the most basic problems in the field of signal processing. The amplitude of frequency reflects the strength of signal energy and its accurate estimation is of great value. Since the 1970s, some scholars have devoted themselves to the research of discrete spectrum correction theory and proposed a number of methods to correct the error of spectrum analysis. The task of spectrum correction is to calculate the frequency, amplitude and phase accurately using the

information provided by discrete spectrum analysis. In order to get the exact position of the frequency, the most direct method is to interpolate values between the amplitude of the main line and several neighboring lines and then calculate the precise frequency position according to an interpolation formula. Such kind of method is called the interpolation spectrum correction methods, such as the ratio method [15,16], the energy center method [17–19], the Candan method [20] and its improved version [21], the Macloed method [22], and the Jacobsen method [23]. These methods are based on the traditional spectrum analysis of fast Fourier transformation (FFT). Therefore, the inherent spectrum leakage will affect the accuracy of spectrum correction. In addition, the accuracy of spectrum correction also suffers from errors of the inter-spectral interference and noise. Spectrum leakage and barrier effect cause the interaction of the various frequencies and then produces errors. Windowing can reduce the spectral interference from each frequency and improve the correction accuracy. Another important impact factor of interpolation spectrum correction methods is the interference from the noise. When there is no noise in the signal, the interpolation spectrum correction methods are accurate, which can correct the frequency, amplitude and phase accurately. However, the correction accuracy decreases when there is a noise, especially when the signal-to-noise ratio (SNR) is small. The spectrum of the noise is always broadband, and the spectrum of the signal is usually narrowband. In actual signals, the spectrum of the noise and signals will overlap and then result in errors. The noise can modify the numerical value of spectral lines and interfere with the location of spectral lines. Carlo offelli and Dario petri studied the effect of the noise on the correction accuracy of the interpolation method [24]. Schoukens analyzed the influence of the noise on the interpolation method qualitatively [25]. Xie Ming, Ding Kang and other scholars also proposed the ratio correction method and developed the interpolation method of the general spectrum correction method, which solved the problem of the accurate solution to the amplitude, phase, and frequency of the discrete spectrum with large frequency interval [15,26–29].

The length of the signal affects the accuracy of the frequency resolution and frequency estimation. The longer the signal is, the higher the frequency resolution will achieve. However, it is not appropriate for real-time processing. The short-time signals make the frequency resolution be limited, which also affects its frequency accuracy. Based on the above analysis, this paper proposes the fault diagnosis method, which combines LMD and the ratio correction method, to improve the accuracy of fault frequencies of rolling bearing, especially for the short-time signals. The feasibility and effectiveness of the proposed method are verified by the analysis of measured signals. This method is also suitable for online real-time monitoring.

## 2. Materials and Methods

### 2.1. LMD and Its Improved Algorithm

Based on LMD, the vibration signal of rolling bearing is decomposed into a series of PFs by cyclic iteration. The algorithm includes a large outer loop and a small inner loop. The inner small loop is assigned to extract the envelope signal and a purely frequency modulated signal which can combine into PFs. The large outer loop is designed to extract PFs from the vibration signal. The specific steps of LMD are described in detail in [5]. After decomposed by LMD, the raw signal $x(t)$ can be reconstructed according to

$$x(t) = \sum_{i=1}^{m} PF_i(t) + u_m(t) \tag{1}$$

where $PF_i(t)$ is PFs, $u_m(t)$ is the residual signal, and $m$ denotes the number of PFs.

LMD also has its own defects, such as sliding step selection, mode mixing, sifting stopping criterion, and end effect. It is essential to improve the algorithm of LMD to make the decomposition convergence, achieve a good decomposition effect, and extract fault feature information more accurate. Some scholars have put forward different methods to solve the problem of sliding step size, such as one third of the longest local mean [5] and cubic spline interpolation method [30]. Other scholars proposed the method of extending the signal on both sides to solve the problem of end effect [31]. Ensemble local

mean decomposition successfully overcomes the mode mixing and can efficiently eliminate various interference contents while preserving fault characteristic information [32]. Related parameters of these methods need to set manually and maybe they are not the best one. Therefore, these methods are not self-adaptive. Self-adaptive sifting stopping criterion builds an objective function, which can determine whether the screening result converges or not, and stop the sifting to ensure the best effect of PFs in the case of the closest theoretical stopping criterion [33]. The algorithm of LMD and self-adaptive sifting stopping criterion are detailed in References [5,33]. The improved algorithm of LMD [33] is used to decompose the vibration signal in this paper.

## 2.2. Ratio Correction Method

In digital signal processing, the sampling signal is a series of finite-length discrete data. Amplitude spectrum of a signal can be gained by the discrete Fourier transform (DFT) or FFT. In other words, the amplitude spectrum of a signal is the result of sampling in the equal frequency domain according to $\Delta\omega = 2\pi/N$ after the convolution of the signal spectrum and a window function. If the frequency of the periodic signal is on a certain spectral line exactly, the frequency, amplitude and phase are accurate. If the frequency is between the two adjacent spectral lines, instead of the main lobe center of the window spectrum, the frequency, amplitude, and phase reflected by the peak spectral line are inaccurate [15].

Spectrum correction is to find the abscissa of the center of the main lobe, as shown in Figure 1a. Assume the spectrum of the window spectrum function is $f(x)$ and its function expression is known. Besides, it is symmetrical about $Y$-axis. $x$ and $x+1$ are adjacent spectral lines and they are closest to the peak of frequency. Their corresponding window spectrum function are $f(x)$ and $f(x+1)$. Their corresponding discrete spectrum are $y_x$ and $y_{x+1}$. $\Delta x = -x$ is the spectral line correction and it can be calculated by the following Equation.

$$\begin{cases} y_x = f(x) \\ y_{x+1} = f(x+1) \end{cases} \tag{2}$$

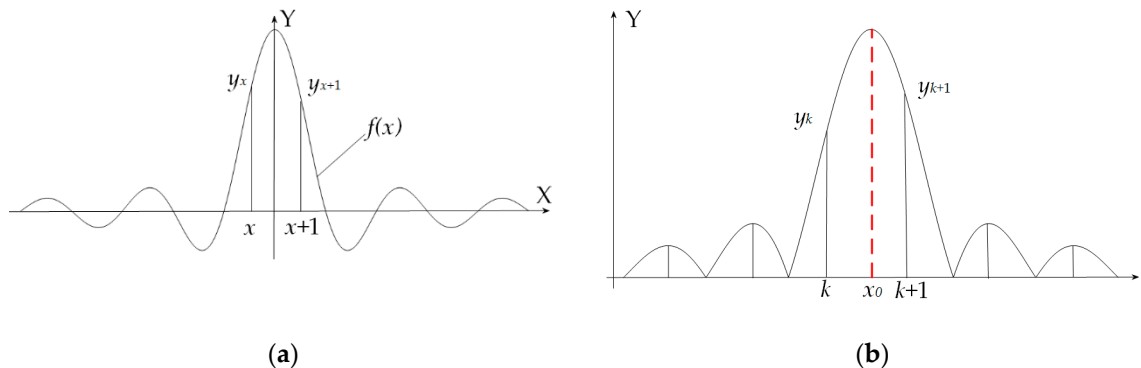

| (a) | (b) |

**Figure 1.** (**a**) Schematic diagram spectrum correction; (**b**) spectrum of window function.

So, constructing a function

$$V = F(x) = \frac{f(x)}{f(x+1)} = \frac{y_x}{y_{x+1}} \tag{3}$$

where $V$ is the ratio of the two spectral lines and it is a function of $x$. The interval between these two spectral lines is 1. The inverse function $V$ is

$$x = g(V) = g\left(\frac{y_x}{y_{x+1}}\right) \tag{4}$$

where the condition for the existence of the inverse function $V$ is that $x$ is in one-to-one correspondence with $V$ in Equation (3). Then, the frequency correction value can be calculated according to $\Delta x = -x$. Therefore, it is called ratio correction method.

In the actual calculation, the main lobe center $x_0$ is at the true frequency of signal, as shown in Figure 1b. $y_k$ and $y_{k+1}$ are two adjacent spectral lines in the main lobe. $k$ and $k+1$ are the serial number of spectral lines. $\Delta x = -x$ is the spectral line correction and it can be calculated by equation

$$\begin{cases} V = \dfrac{y_k}{y_{k+1}} \\ x = g(V) \end{cases} \tag{5}$$

Then, the correction frequency formula is as following

$$f_k = \frac{(k + \Delta x)F_S}{N} \tag{6}$$

where $F_s$ is the sampling frequency and $N$ is the sampling number.

### 2.3. Diagnostic Method Flow and Simulation Analysis

For the short-time signals of rolling bearing, the proposed method of this paper, which combines LMD and the ratio correction method, is suitable for the online fast diagnosis. Firstly, the improved LMD decomposes the raw signal into a set of PFs and the residual signal. The PF, which contains most of the fault information, is selected by the correlation coefficients $\rho$ between the raw signal and PFs. The correlation coefficient $\rho$ is a statistical index that describes the degree of correlation between two variables. Its values range from $-1$ to 1. The closer the absolute value of $\rho$ is to 1, the stronger the correlation between two variables is; the closer the absolute value of $\rho$ is to 0, the weaker the correlation between two variables is. The formula for the correlation coefficient $\rho$ of the discrete signals $m_i$ and $n_i$ is as following

$$\rho = \frac{\sum\limits_{i=1}^{N} (m_i - \overline{m})(n_i - \overline{n})}{\sqrt{\sum\limits_{i=1}^{N} (m_i - \overline{m})^2 \sum\limits_{i=1}^{N} (n_i - \overline{n})^2}} \tag{7}$$

where $N$ is the number of $m_i$ or $n_i$. Secondly, the Hilbert envelope spectrum of the selected PF is calculated by HT and spectrum analysis. Additionally, the ratio correction method is then used to correct the Hilbert envelope spectrum of the selected PF. Finally, higher precision fault frequencies are extracted and the fault location is accurately determined. In order to highlight the advantages of the proposed method in processing the short-time signals, this paper makes a comparative analysis between the long-time signals and the short-time signals. The specific process of the proposed method is shown in Figure 2.

## 3. Application and Results

To verify the effectiveness of the proposed method, the rolling bearing data disclosed by the Case Western Reserve University Bearing Data Center are used [34]. As shown in Figure 3, the test stand consists of a 2 hp motor (left), a torque transducer/encoder (center), a dynamometer (right), and control electronics. The motor speeds are from 1797 r/min to 1720 r/min. The test bearings support the motor shaft. Single point faults were introduced to the test bearings separately at the inner raceway, rolling element and outer raceway using electro-discharge machining with fault diameters of 7 mils, 14 mils, 21 mils, 28 mils, and 40 mils (1 mil = 0.001 inches). The test bearing is a deep groove ball bearing of SKF6205 and its related parameters are as follows: the inner diameter is 25 mm, the outer diameter is 52 mm, the number of balls is 9, the rolling body diameter is 7.94 mm, the pitch diameter is 39.04 mm, and the contact angle is 0°. The vibration data were collected by accelerometers mounted on the

housing with magnetic bases. Accelerometers were placed at the 12 o'clock position at both the drive end and fan end of the motor housing. The vibration signals were collected using a 16 channel DAT recorder, and were post processed in a MATLAB environment. The sampling frequency is 12 kHz. The fault data at the drive end with the smallest diameter (7 mils) at the inner raceway and the outer raceway are selected in this paper. The outer raceway faults were located at 3 o'clock. The rotation speed (RS), the rotation frequency (RF), the inner raceway fault frequency (IRF), and the outer raceway fault frequency (ORF) are shown in Table 1.

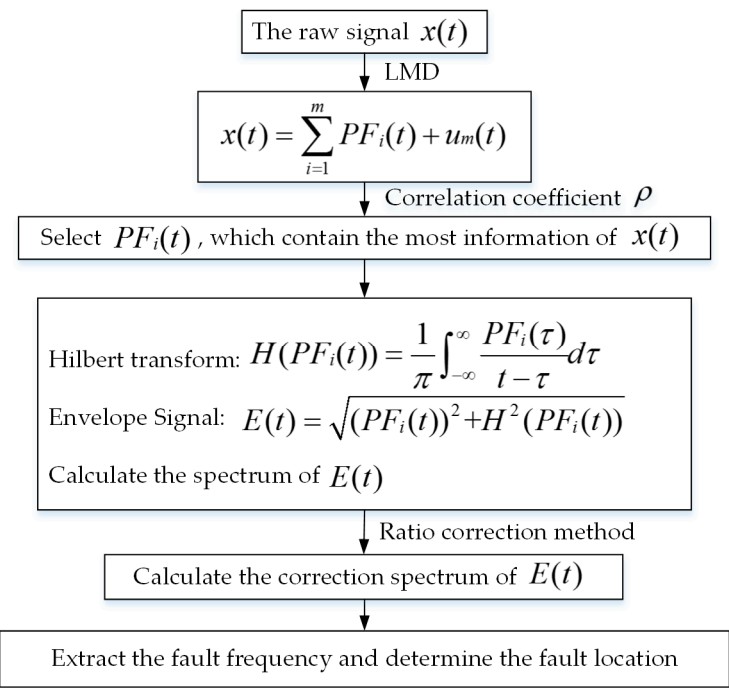

**Figure 2.** The specific process of diagnostic method.

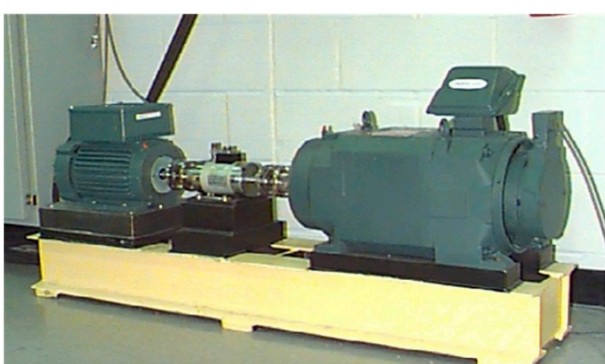

**Figure 3.** The test stand for normal and faulty test data of ball bearings.

**Table 1.** The fault frequency of inner raceway and outer raceway, where $x_{in}$ is the inner raceway fault data, and $x_{out}$ is the outer raceway fault data.

| Data Code | RS | RF | IRF | ORF |
|---|---|---|---|---|
| $x_{in}$ | 1721 r/min | 28.68 Hz | 155.3 Hz | |
| $x_{out}$ | 1725 r/min | 28.75 Hz | | 103.1 Hz |

### 3.1. Fault Diagnosis of Bearing with Fault at the Inner Raceway

The time-domain waveform of the fault signal $x_{in}$ from the inner raceway is shown in Figure 4a, where the sampling number of $x_{in}$ is 12000. Divide the signal $x_{in}$ into four segments of $x_{in1}$, $x_{in2}$, $x_{in3}$,

and $x_{in4}$. The number of each segment is 2048. Figure 4b displays the time-domain waveform of $x_{in1}$. LMD is used to decompose $x_{in}$ and $x_{in1}$ separately into a series of PFs and $u_m(t)$ is the residual component, as shown in Figure 5. The correlation coefficients $\rho$ between the raw signal and its PFs were calculated, as shown in Table 2. It is obvious that both of $PF_1(t)$ contain the richest fault information.

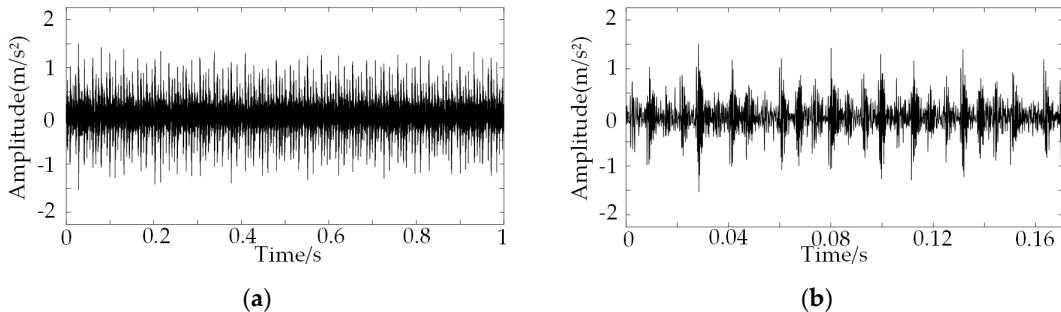

|       |       |
|-------|-------|
| **(a)** | **(b)** |

**Figure 4.** Time-domain waveforms of $x_{in}$ and $x_{in1}$ with different sampling number: (**a**) The sampling number of $x_{in}$ is 12,000; (**b**) the number of first segment $x_{in1}$ is 2048.

**Table 2.** The correlation coefficients $\rho$: $\rho_{in}$ are the correlation coefficients between $x_{in}$ and its PFs; $\rho_{in1}$ are the correlation coefficients between $x_{in1}$ and its PFs.

| $\rho$ | $PF_1(t)$ | $PF_2(t)$ | $PF_3(t)$ | $PF_4(t)$ | $PF_5(t)$ | $PF_6(t)$ |
|--------|-----------|-----------|-----------|-----------|-----------|-----------|
| $\rho_{in}$ | 0.9966 | 0.18162 | 0.0202 | 0.0004 | −4.42E−5 | 2.66E−5 |
| $\rho_{in1}$ | 0.9770 | 0.2096 | 0.0208 | 0.0019 | | |

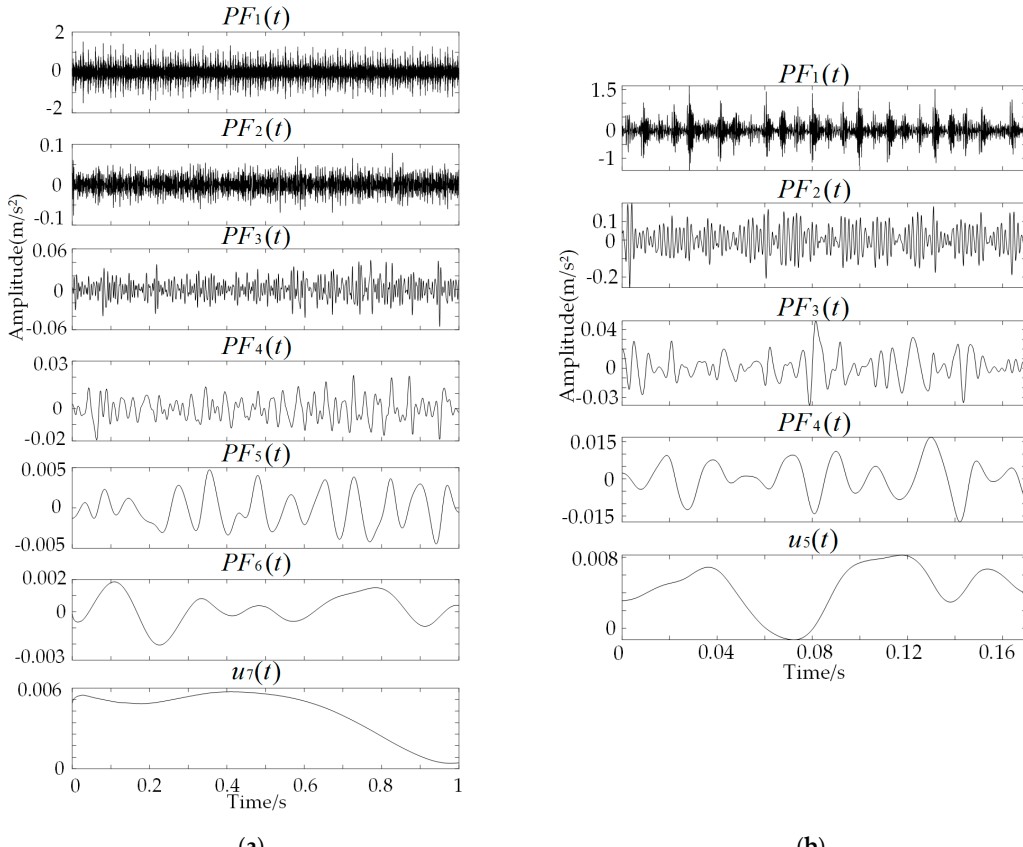

|       |       |
|-------|-------|
| **(a)** | **(b)** |

**Figure 5.** PFs of $x_{in}$ and $x_{in1}$ with different sampling number: (**a**) The sampling number of $x_{in}$ is 12000; (**b**) the number of first segment $x_{in1}$ is 2048.

To raise the effect of contrast, the comparison chart of the Hilbert envelope spectrum of $PF_1(t)$ which is from Figure 5a is shown in Figure 6. The fault frequency 154.6 Hz, which is not corrected, is basically equal to the theoretical fault frequency 155.3 Hz, as shown in Figure 6a. The double frequency 309.9 Hz and the triple frequency 464.4 Hz exist obviously. Similarly, the fault frequency 154.9 Hz, which is corrected, is basically equal to the theoretical fault frequency 155.3 Hz, as shown in Figure 6b. The double frequency 309.8 Hz and the triple frequency 465.1 Hz exist also obviously. In this case, whether it is corrected or not, the fault frequency is almost equal to the theoretical fault frequency. In other words, for the long-time signals, it does not affect the accurate extraction of fault frequency whether the fault frequency is corrected or not.

However, for the short-time signals, there is a difference between the fault frequency and the theoretical fault frequency. As shown in Figure 7a, there is a clear error of 2.8 Hz between the fault frequency 152.5 Hz and the theoretical fault frequency 155.3 Hz. The fault frequency 155 Hz, which is the value of spectrum correction, is almost equal to the theoretical fault frequency 155.3 Hz, as shown in Figure 7b. After calculating the remaining three segments of data $x_{in2}$, $x_{in3}$, and $x_{in4}$, the comparison chart of the Hilbert envelope spectrum of $PF_1(t)$ is shown in Figure 8. Without spectrum correction, the fault frequency 152.5 Hz has an error according to the theoretical fault frequency 155.3 Hz, as shown in Figure 8a. With spectrum correction, the fault frequency 155 Hz and 154.9 Hz are almost equal to the theoretical fault frequency 155.3 Hz, as shown in Figure 8b. The noise and spectral intensity of each segment are different from each other, which affects the value of the spectrum correction, as shown in Figure 8b. However, it does not affect the accurate extraction of the fault frequency. Through spectrum correction, the sampling number of signals can be shortened, while keeping an accurate frequency value.

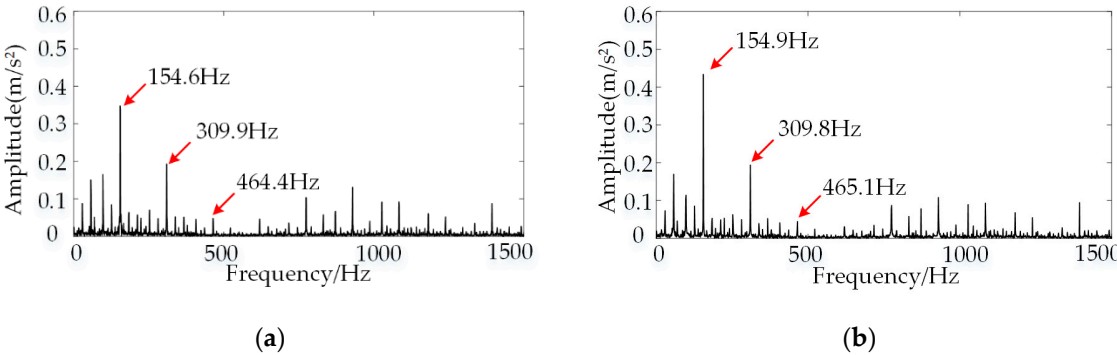

**Figure 6.** Comparison chart of the Hilbert envelope spectrum of $PF_1(t)$ which is from Figure 5a: (**a**) Without spectrum correction; (**b**) with spectrum correction.

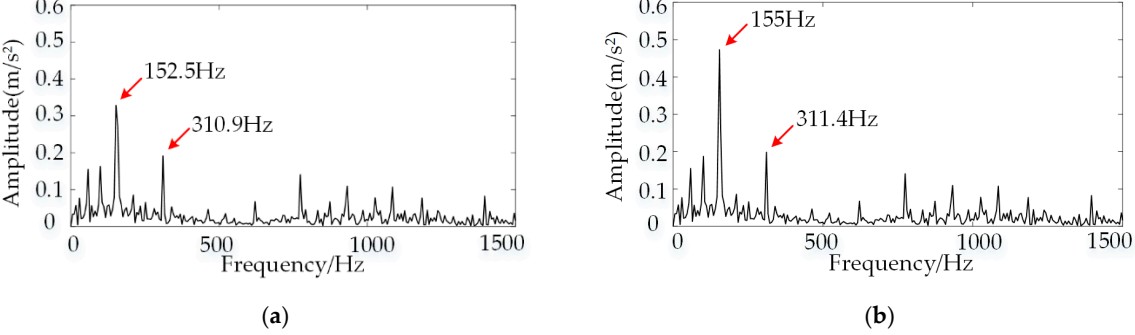

**Figure 7.** Comparison chart of the Hilbert envelope spectrum of $PF_1(t)$ which is from Figure 5b: (**a**) Without spectrum correction; (**b**) with spectrum correction.

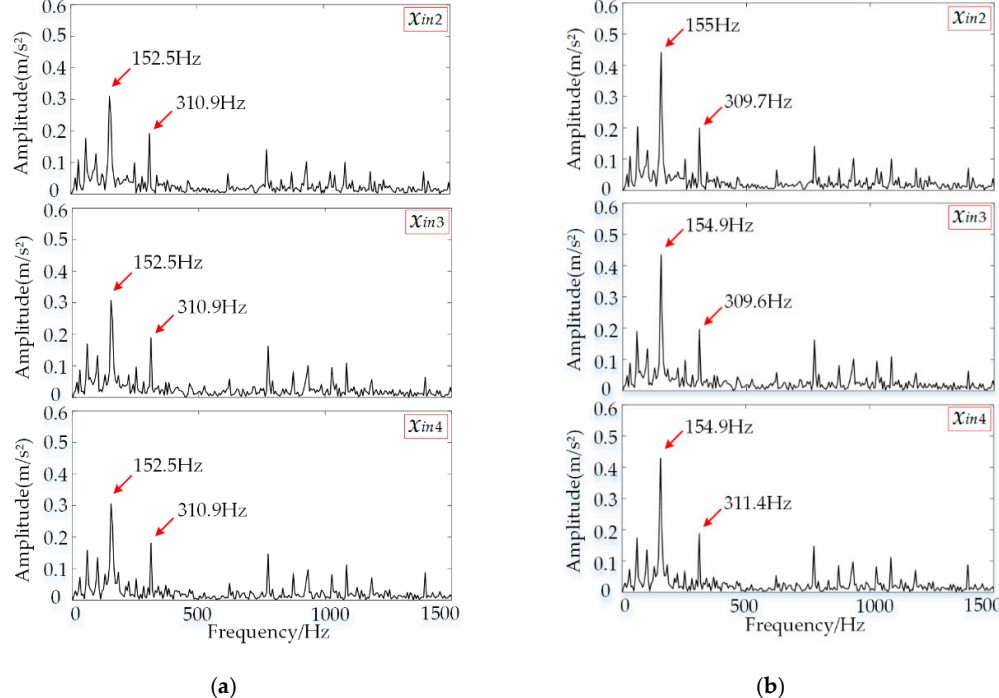

**Figure 8.** Comparison chart of the Hilbert envelope spectrum of $PF_1(t)$ after calculating the remaining three segments of data: (**a**) Without spectrum correction; (**b**) with spectrum correction.

### 3.2. Fault Diagnosis of Bearing with Fault at the Outer Raceway

The time-domain waveform of the fault signal $x_{out}$ from the outer raceway is shown in Figure 9a, where the sampling number is 12000. Divide the signal $x_{out}$ into four segments of $x_{out1}$, $x_{out2}$, $x_{out3}$, and $x_{out4}$. The number of each segment is 2048. Figure 9b displays the time-domain waveform of $x_{out1}$. LMD is used to decompose $x_{out}$ and $x_{out1}$ separately into a series of PFs and $u_m(t)$ is the residual component, as shown in Figure 10. The correlation coefficients $\rho$ between the raw signal and its PFs were calculated, as shown in Table 3. Similarly, it is obvious that both of $PF_1(t)$ contain most fault information.

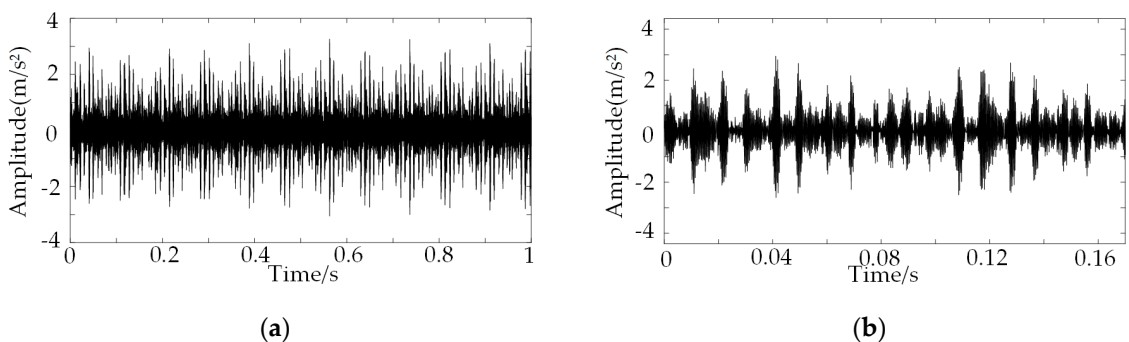

**Figure 9.** Time-domain waveforms of $x_{out}$ and $x_{out1}$ with different sampling number: (**a**) The sampling number of $x_{out}$ is 12,000; (**b**) the number of first segment $x_{out1}$ is 2048.

**Table 3.** The correlation coefficients $\rho$: $\rho_{out}$ is the correlation coefficients between $x_{out}$ and its PFs; $\rho_{out1}$ is the correlation coefficients between $x_{out1}$ and its PFs.

| $\rho$ | $PF_1(t)$ | $PF_2(t)$ | $PF_3(t)$ | $PF_4(t)$ | $PF_5(t)$ | $PF_6(t)$ |
|---|---|---|---|---|---|---|
| $\rho_{out}$ | 0.9986 | 0.0602 | 0.0106 | 0.0007 | 0.0001 | 3.605E−5 |
| $\rho_{out1}$ | 0.9992 | 0.0380 | 0.0090 | 0.0005 | | |

The comparison chart of the Hilbert envelope spectrum of $PF_1(t)$ which is from Figure 10a is shown in Figure 11. The fault frequency 103.3 Hz, which is not corrected, is basically equal to the theoretical fault frequency 103.1 Hz, as shown in Figure 11a, and the double frequency 207.3 Hz exists obviously. Similarly, the fault frequency 103.5 Hz, which is corrected, is almost equal to the theoretical fault frequency 103.1 Hz, as shown in Figure 11b, and the double frequency 207 Hz exists also obviously. Whether the fault frequency is corrected or not, it is almost equal to the theoretical fault frequency, but the corrected fault frequency is farther from the theoretical fault frequency than the uncorrected fault frequency. The reason is that the theoretical solution to the ratio correction method is affected by the noise. When the SNR is low to a certain extent, the spectral line of the spectrum will be overwhelmed by the noise, which may cause an initial positioning error on the signal [35]. Without the noise between the frequencies, the algorithm of the ratio correction method is simple and its calculation accuracy is high. However, the key to the correction accuracy is to find the most maximum lines of $y_k$ and $y_{k+1}$ accurately in the main lobe, as shown in Figure 1b, and then get the correction frequency according to the Equation (6). If the SNR is relatively high, the two maximum lines in the main lobe can always be found correctly. When SNR is low to some extent, the amplitude of the maximum spectral line becomes small. It is easy to find a sub-largest line in the opposite direction and then result in a wrong line correction, which will reduce the accuracy of the correction [36]. Due to the interference of the noise or frequency leakage, even in the case of high SNR, there are some cases that the true spectral line may be wrongly selected [24,37]. In this regard, a more in-depth research is needed. However, when analyzing the long-time signals, it does not affect the accuracy of the fault frequency whether the fault frequency is corrected or not.

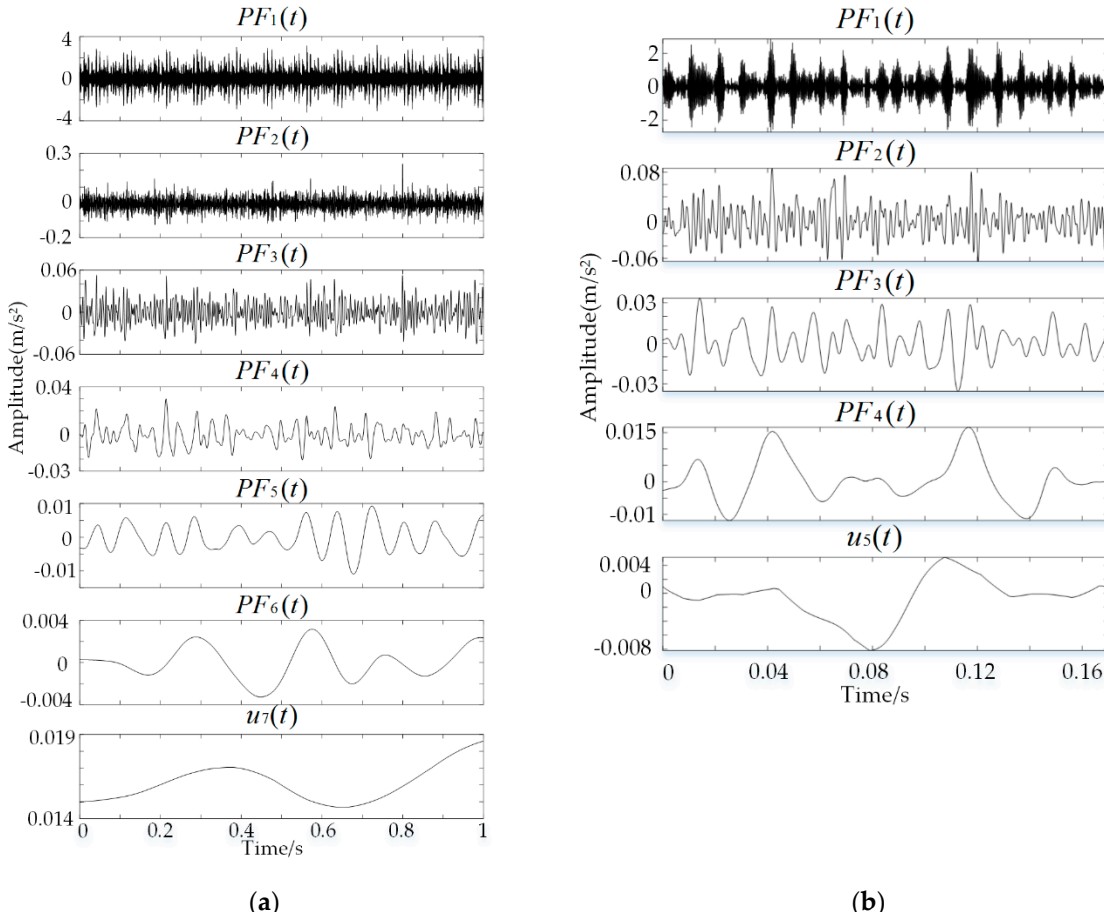

(**a**)　　　　　　　　　　　　　　　　　　　　　　　　　　　　　(**b**)

**Figure 10.** PFs of $x_{out}$ and $x_{out1}$ with different sampling number: (**a**) The sampling number of $x_{out}$ is 12,000; (**b**) the number of first segment $x_{out1}$ is 2048.

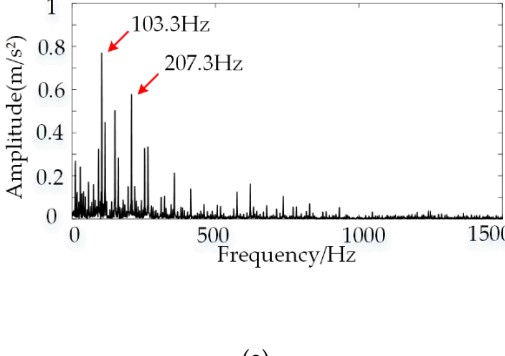

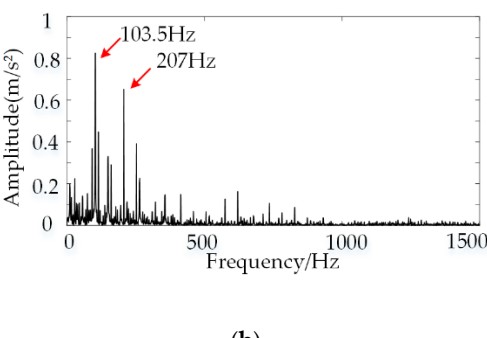

(**a**)    (**b**)

**Figure 11.** Comparison chart of the Hilbert envelope spectrum of $PF_1(t)$ which is from Figure 10a: (**a**) Without spectrum correction; (**b**) with spectrum correction.

For the short-time signals, there is an error between the uncorrected fault frequency and the theoretical fault frequency. As shown in Figure 12a, the fault frequency 105.6 Hz has a difference of 2.5 Hz according to the theoretical fault frequency 103.1 Hz. However, the fault frequency 103.3 Hz, which is corrected, is almost equal to the theoretical fault frequency 103.1 Hz, as shown in Figure 12b. The comparison chart of the Hilbert envelope spectrum of $PF_1(t)$ is shown in Figure 13, where the remaining three segments of data $x_{out2}$, $x_{out3}$, and $x_{out4}$ are calculated.

Without spectrum correction, the fault frequency 105.6 Hz has a clear error according to the theoretical fault frequency 103.1 Hz, as shown in Figure 13a. That is because the frequency resolution $\Delta w$ is related to the sampling frequency $F_s$ and the sampling number $N$. The relationship between them is expressed by the formula of $\Delta w = F_s/N$. When the sampling frequency is constant, the frequency resolution $\Delta w$ is inversely proportional to the sampling number $N$. When the number of the four segments is 2048, the frequency resolution decreases and the spectrum is distorted. However, with spectrum correction, the fault frequency 103.4 Hz is almost equal to the theoretical fault frequency 103.1 Hz, as shown in Figure 13b. The spectrum correction method can correct the spectral distortion of the short-time signals and restore the authenticity of the spectrum to some extent.

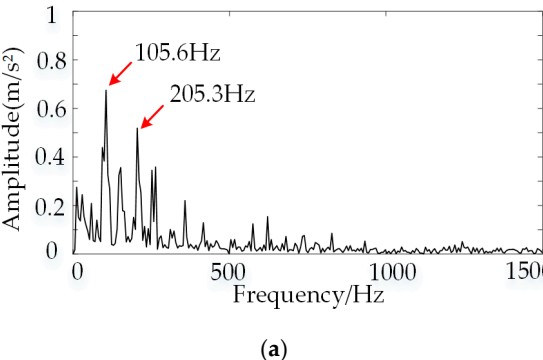

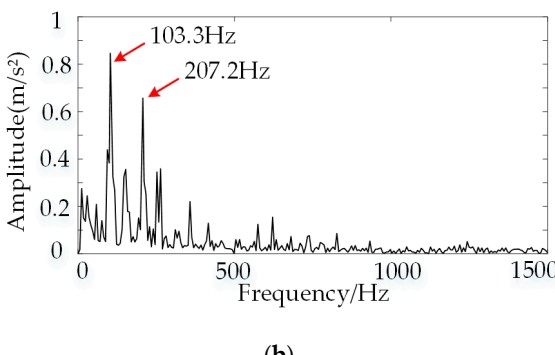

(**a**)    (**b**)

**Figure 12.** Comparison chart of the Hilbert envelope spectrum of $PF_1(t)$ which is from Figure 10b: (**a**) Without spectrum correction; (**b**) with spectrum correction.

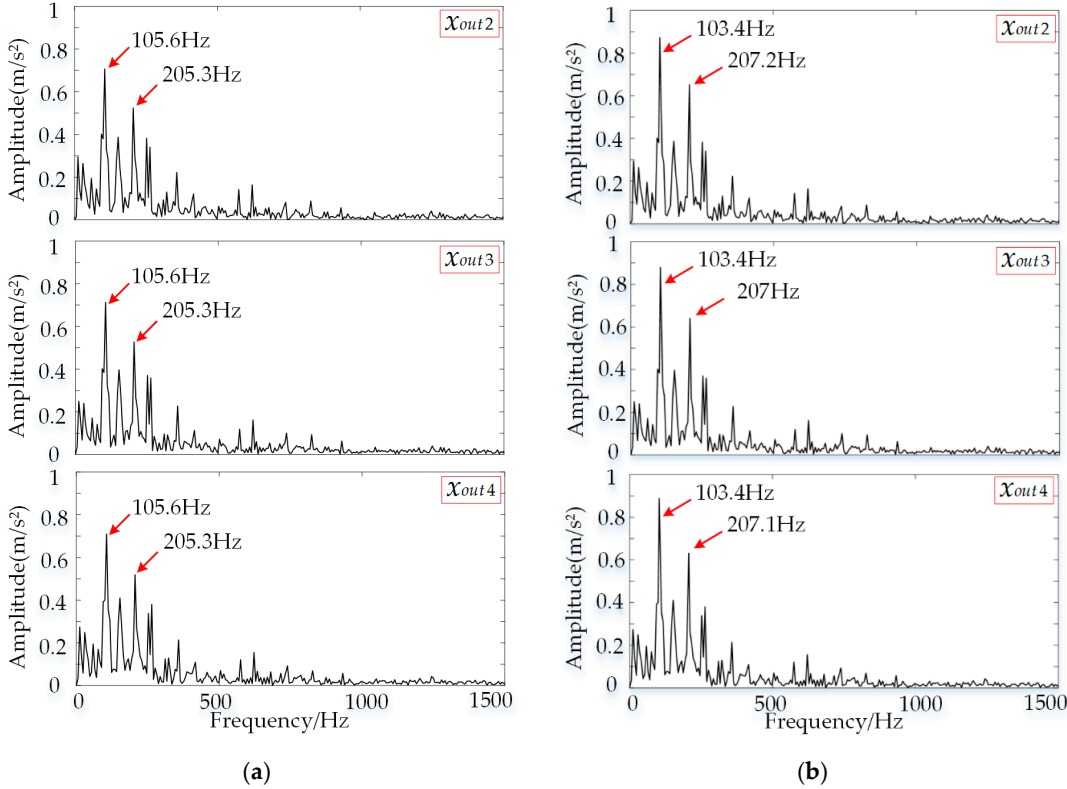

**Figure 13.** Comparison chart of the Hilbert envelope spectrum of $PF_1(t)$ after calculating the remaining three segments of data: (**a**) Without spectrum correction; (**b**) with spectrum correction.

Take 20 inner raceway fault samples to form the sample sets $S_{in}$. The number of each sample of 10 samples is 12,000, and the number of each sample of the other 10 samples is 2048. Similarly, take 20 outer raceway fault samples to form the sample sets $S_{out}$. The number of each sample of 10 samples is 12,000, and the number of each sample of the other 10 samples is 2048. Their IRF and ORF are shown in Table 1. The fault frequencies were extracted from $S_{in}$ and $S_{out}$, as shown in Figures 14 and 15. In the Figures 14 and 15, WOSC denotes the fault frequency without spectrum correction and WSC denotes the fault frequency with spectrum correction. As can be seen from Figures 14a and 15a, for the long-time signals, there is very small error between the corrected fault frequency or the uncorrected fault frequency and the theoretical fault frequency. The very small error does not affect the accurate extraction of fault frequency. As can be seen from Figures 14b and 15b, for the short-time signals, there is a large error between the uncorrected fault frequency and the theoretical fault frequency, or the corrected fault frequency. The large error may affect the accurate extraction of fault frequency.

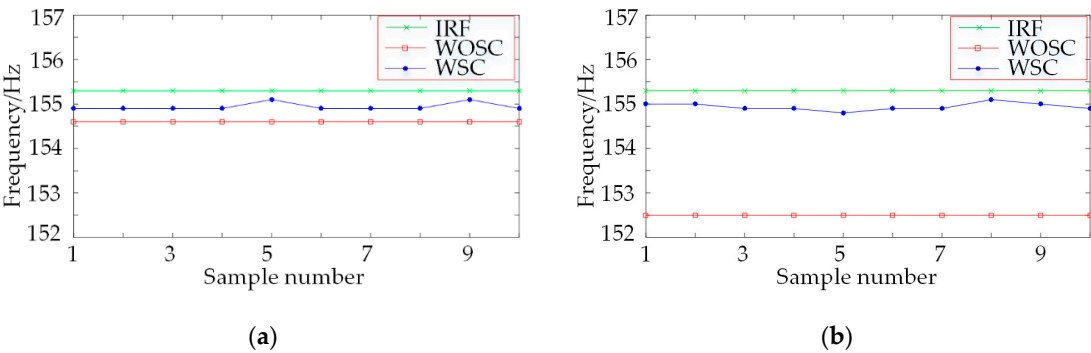

**Figure 14.** The sample sets $S_{in}$: (**a**) The sampling number is 12,000; (**b**) the sampling number is 2048.

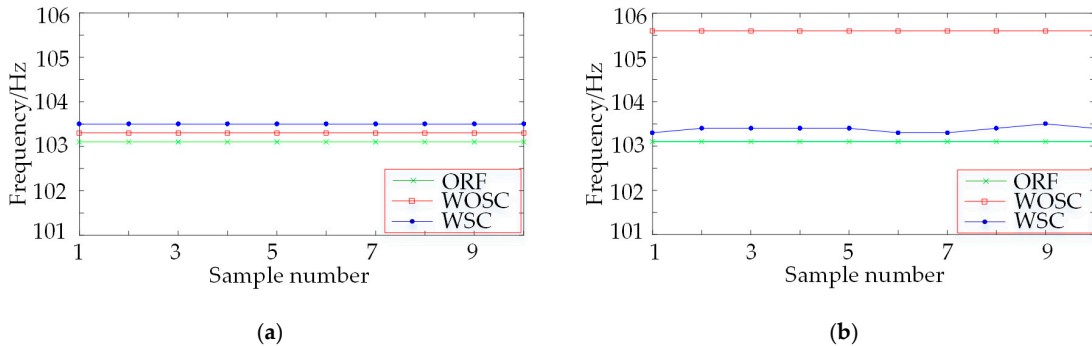

**Figure 15.** The sample sets $S_{out}$: (**a**) The sampling number is 12,000; (**b**) the sampling number is 2048.

## 4. Discussion

The fault signal of rolling bearing typically has non-stationary, nonlinear, and modulated characteristics. Compared with traditional signal processing technology, LMD can decompose signals self-adaptively on the basis of signals themselves. For the online real-time monitoring, it is an effective way to increase the diagnostic efficiency by reducing the sampling data while keeping the accuracy of fault frequency. For the long-time signals of rolling bearing, the fault frequency is almost equal to the theoretical fault frequency whether it is corrected or not. Therefore, it does not affect the extraction accuracy of fault frequency. However, for the short-time signals, the uncorrected fault frequency has a certain error compared to the theoretical fault frequency. However, with spectrum correction, the fault frequency is basically equal to the theoretical fault frequency. It is shown that the proposed method of this paper is valid.

However, the proposed method of this paper has some shortcomings. How to properly determine the sampling number of the short-time signals in the case of ensuring frequency accuracy and computational efficiency is a problem. It can be manually set according to the short-time signals itself. However, the signal is changing with the environment. If the sampling number does not change with the change of the signal, it may lead to the mistakes of diagnosis, especially in the online real-time monitoring. Another problem is how to completely eliminate the influence of the noise to the ratio correction method. The noise not only affects the direction of interpolation, but also affects the spectral accuracy. How to avoid the interpolation in the opposite direction and get the determined correction frequency is another problem to be solved. Future research will focus on the further improvement of the algorithm and the coalescent of the proposed method of this paper and deep learning.

## 5. Conclusions

The fault diagnosis method which uses LMD and the ratio correction method is proposed for the short-time signals of rolling bearing. The fault signal of rolling bearing at the inner raceway and the outer raceway were analyzed by the proposed method. The results show that this method can gain a high frequency resolution, and extract the fault frequency accurately under the condition of the short-time signals. Therefore, this method has a certain value of engineering applications. The proposed method of this paper can reduce the size of data samples while ensuring accuracy. The deep learning models can directly work on the raw data without any data preprocessing. Thus, it is worthwhile to make further study of combining spectrum correction with the deep learning models for processing the short-time signals. In earthquake engineering, many methods are utilized to solve problems related to earthquakes. For example, probabilistic risk-based performance evaluation of seismically was used to analyze base-isolated steel structures under the influence of far-field earthquakes [38]. Besides, signal processing is one of the most typical applications in earthquake engineering. Seismic signals have the characteristics of non-stationary and "short-time" random impulses. Thus, it is worthwhile to make further study for handling the short-time signals of earthquake data by the proposed method of this paper.

**Author Contributions:** Conceptualization, Y.D. and C.W.; methodology, Y.D.; software, Y.D. and C.W.; validation, Y.D. and C.W.; formal analysis, Y.D., C.W. and P.L.; investigation, Y.C.; resources, Y.C.; data curation, Y.D., C.W. and P.L.; writing—original draft preparation, Y.D.; writing—review and editing, Y.D., Y.C. and C.W.; visualization, C.W.; supervision, C.W.; project administration, C.W.; funding acquisition, C.W.

**Funding:** This research was funded by the National Key R & D Plan Program of China (2018YFB0106100), the Sichuan Science and Technology support Program (2019YFG0352, 2019YFG0098, 2017GZ0395 and 2017GZ0394).

**Conflicts of Interest:** The authors declare no conflict of interest.

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
