# Peer review of "Improving the Accuracy of Fault Frequency by Means of Local Mean Decomposition and Ratio Correction Method for Rolling Bearing Failure"

_applsci, doi:10.3390/app9091888_

Round 1
Reviewer 1 Report
Paper is very good represented, starting from Introduction, representation of others work. Representation of methods and results is very clear. Also are included comparisons of results of other authors.
Author Response
Many thanks for your postive comments!
Reviewer 2 Report
See attached

Author Response
1.Your manuscript aims to deal with a real-time monitoring used in fault diagnosis. This interesting topic has been under investigation for several years, where multiple approaches have been proposed. However, your document is missing some of them and you should address them in the introduction. Please be aware that you should not only refer to them but also compare their (dis)advantages to your proposed framework. Specifically, I propose to review the following papers and add them in the introduction:
a. Wang, L., Liu, Z., Miao, Q., & Zhang, X. Time–frequency analysis based on ensemble local mean 461 decomposition and fast kurtogram for rotating machinery fault diagnosis. Mechanical Systems and Signal 462 Processing, 2018, 103, 60-75.
b. Jing, L., Zhao, M., Li, P., & Xu, X. A convolutional neural network based feature learning and fault 476 diagnosis method for the condition monitoring of gearbox. Measurement, 2017, 111, 1-10.
c. Yang, Y., Liao, Y., Meng, G., & Lee, J. A hybrid feature selection scheme for unsupervised learning and its 492 application in bearing fault diagnosis. Expert Systems with Applications, 2011, 38(9), 11311-11320.
d. Pan, H., He, X., Tang, S., & Meng, F. An Improved Bearing Fault Diagnosis Method using 501 One-Dimensional CNN and LSTM. Strojniski Vestnik/Journal of Mechanical Engineering, 2018, 64.
Response: Many thanks for your comments! I read these documents in detail and cited them in the revised manuscript. At the same time, we compared the disadvantages and advantages of these documents to the proposed framework of my paper.
2. Shortly, how can you justify your proposed methodology against the application of convolutional neural networks with first-layer kernels, using the normalized Fourier coefficients, which are transformed from the raw temporal signals?
Response: Many thanks for your comments! The proposed method of our paper does not mention the application of neural networks. Please tell me the specific instructions.
3. I would like to see the advantage and disadvantage of your proposed methodology against the deep learning models, which can directly work on the raw data without any data preprocessing. Try to include a sentence out of that in your conclusion.
Response: Many thanks for your comments! In the revised manuscript, we have added the advantage and disadvantage of the proposed method against the deep learning models in the introduction and conclusion.
4. The last paragraph of the Section 3.2, lines 288-289: “Without spectrum correction, the analyzed frequency of 105.6Hz has a large error according to the theoretical fault frequency of 103.1Hz, as shown in Figure 12(a).” Why? Explain the reason behind of that.
Response: Many thanks for your comments! In the revised manuscript, we have added the reason in the paper. The reason is as following:
That is because the frequency resolution is related to the sampling frequency and the sampling number . The relationship between them is expressed by the formula of . When the sampling frequency is constant, the frequency resolution is inversely proportional to the sampling number . When the number of the four segments is 2048, the frequency resolution decreases and the spectrum is distorted.
5. In line 260, you wrote “When the SNR is low to a certain extent, the spectral line in the spectrum will be overwhelmed by the noise, which may cause an initial positioning error on the signal [30]”. Is it because of this fact that the theoretical solution of ratio correction method is affected by noise? If not, please justify your claim by providing reasons.
Response: Many thanks for your comments! the theoretical solution of ratio correction method is affected by noise and inter-spectral interference. This has been discussed in the introduction. The discussion is as following in the revised manuscript :
the accuracy of spectrum correction is also affected by inter-spectral interference error and noise interference error. Spectrum leakage and barrier effect cause the interaction of the various frequencies in the signal, and then produces errors.
When there is no noise in the signal, the interpolation spectrum correction methods are accurate, which can correct the frequency, amplitude, and phase of the signal accurately, but the correction accuracy will decrease when there is noise, especially when the signal-to-noise ratio (SNR) is small. The spectrum of noise is always broadband, and the spectrum of the signal is usually narrowband. In actual signals, the spectrum of noise and signals will overlap and then results in error. Noise can modify the numerical value of spectral lines and interfere with the location of spectral lines.
6. The conclusion must be improved through adding this comment, which is missing in your paper. The proposed methodology of yours is capable to take the online real-time monitoring technology into account. Specially, the proposed algorithm should be efficient for short-time signals. One of the most important area of application of this is Earthquake Engineering, where the effect of “short-time” random impulses is investigated. The conclusion section is missing an outlook which opens the application of your work in the mentioned field. This is directly correlated to the “applied science”. As such, the authors should put couple of sentences in their conclusion part, talking about the “possibility” of their method in being applied in “earthquake engineering”. Authors should use the following recent paper, as the references in earthquake engineering, and should address them in their text and list of references:
a. Rezaei Rad A, Banazadeh M. Probabilistic risk-based performance evaluation of seismically base-isolated steel structures subjected to far-field earthquakes. Buildings 2018;8. doi:10.3390/buildings8090128.
Response: Many thanks for your comments! we added an outlook about the “possibility” of proposed method of this paper in being applied in “earthquake engineering” and cited the references about earthquake engineering in the revised manuscript.
7. In the discussion section of your manuscript, the weakness of your method is somehow biased and you should be able to put commentary about the weaknesses of the proposed method.
Response: Many thanks for your comments! We have added the weakness of the proposed method in the discussion section of the revised manuscript.
8. Line 197-198, where it is written “The defect of the bearing is a single point damage of electric discharge machining”; the sentence is vague and you try to clarify the sentence.
Response: Many thanks for your careful check! We are sorry for our negligence! We added the details of the experiment in the Application and Results section.
Single point faults were introduced to the test bearings separately at the inner raceway, rolling element and outer raceway using electro-discharge machining with fault diameters of 7 mils, 14 mils, 21 mils, 28 mils, and 40 mils (1 mil=0.001 inches).
9. The authors should review the text again and be sure you have used the appropriate articles, “the”, “a” and “an”. There were lots of mistakes in that regard.
Response: Many thanks for your careful check! We are sorry for our negligence. We removed lots of mistakes about “the”, “a” and “an”, and made the article more fluent.
10. Overall, you should keep a same verb tense for each section. The literature-review section in past, the methodology and research outcome in the present, and the conclusion in the past form.
Response: Many thanks for your careful check! We are sorry for our negligence. We removed some grammatical errors about verb tense and modified some sentences to make the article more fluent.
11. The manuscript should be revised with respect to the punctuation rules.
Response: Many thanks for your careful check! We are sorry for our negligence. We revise the manuscript with respect to the punctuation rules.
12. In case the paper length does not become an issue, it is recommended to include a description on how the faults were generated. This might also include photographs of the damaged bearings (this comment is more optional and I am just proposing it).
Response: Many thanks for your careful check! Fault data from the Case Western Reserve University Bearing Data Center. The description on how the faults were generated was added in the Application and Results section.
13. In my opinion, the sampling numbers can play a very important role. Therefore, you should describe the sample sets, as their quality and richness are vital for the final results.
Response: Many thanks for your comments! We have added the description of the sample sets in the Application and Results section of the revised manuscript.
14. The word “Frequencys” you used in the title does not mean anything in English.
Response: Many thanks for your careful check! We are sorry for our negligence. We have modified this word in the title of the revised manuscript.
15. Line 66, the reference source cannot be found and you should modify that
Response: Many thanks for your careful check! We are sorry for our negligence. We have modified this mistakes in the revised manuscript.
16. Make Figure 4 larger. The content cannot be well understood. The same is also applied for Figure 9
Response: Many thanks for your careful check! We are sorry for our negligence. We have modified this mistakes in the revised manuscript.
17. Please note that the nomenclature of all the mathematical symbols should be reflected in your paper.
Response: Many thanks for your careful check! We are sorry for our negligence. We have modified this mistakes in the revised manuscript.
18. The units for speed, and frequency should be added to Table 1.
Response: Many thanks for your careful check! We are sorry for our negligence. We have modified this mistakes in the revised manuscript.
19. In the title, put a space between “LMD” and “and ration …”, in the current version it is: “LMDand Ration ..”
Response: Many thanks for your careful check! We are sorry for our negligence. We have modified this mistakes in the revised manuscript.
20. Define LMD in the title.
Response: Many thanks for your careful check! We are sorry for our negligence. We have modified this mistakes in the revised manuscript.

Reviewer 3 Report
The authors proposed the local mean decomposition and ratio correction methodologies for short-time signals to diagnose a fault in ball bearings. I have the following comments:
1- The authors have given a thorough introduction to the problem including all the current methodologies.
2- The solution is presented in a clear way especially in derivation of
equations 1-5.
3- In the specific process of diagnostic method table (section 2.4), the methodology is given without mathematical formulations for each step. It would be more solid if there are equations associated with each step.
4-The results and experiments are presented well.
5- I agree with the authors that the LMD method should be used with AI methods for diagnosing purposes.
Errata:
Page 2 Line 66: It should mention the reference 10.
Page 4 Line 152: The dot over X should be removed.
Author Response
Reviewer 3
Comments:
1. The authors have given a thorough introduction to the problem including all the current methodologies.
Response: Many thanks for your comments!
2. The solution is presented in a clear way especially in derivation of equations 1-5.
Response: Many thanks for your comments!
3. In the specific process of diagnostic method table (section 2.4), the methodology is given without mathematical formulations for each step. It would be more solid if there are equations associated with each step.
Response: Many thanks for your comments! We are sorry for our negligence. We have added the equations associated with each step in the revised manuscript.
4. The results and experiments are presented well.
Response: Many thanks for your comments!
5. I agree with the authors that the LMD method should be used with AI methods for diagnosing purposes.
Response: Many thanks for your comments!
Errata:
Page 2 Line 66: It should mention the reference 10.
Response: Many thanks for your careful check! We are sorry for our negligence. We have modified this mistakes in the revised manuscript.
Page 4 Line 152: The dot over X should be removed.
Response: Many thanks for your careful check! We are sorry for our negligence. We have modified this mistakes in the revised manuscript.
Reviewer 4 Report
The paper deals with
Improving the Accuracy of Fault Frequencies by Means of LMDand Ratio Correction Methods for Rolling Bearing Failure. In spite of the incontestable appeal of the proposed approach, there are several points, which impair its quality:
1) First of all, the authors should avoid using uncommon shortcuts in the title.
2) Secondly, making an orthographic error in the title is not a good idea (Frequencies).
3) The fault frequencies are as they are and cannot be improved. One can only improve its estimation quality.
4) The above comments should be also incorporated within the abstract.
5) What is a particular meaning of “1” in eq. 1?
6) The authors should provide the existence condition of an inverse (3) (Improve equation quality as well).
7) Delata x and Delat k denote the same thing. The authors should condense such a description.
8) What is the meaning of * in (5)?
9) The authors introduce modulation strategies without any prior background. The reader will be confused while analysing this part of the paper.
10) The main contribution of the paper (Sec. 2.4) has just a few lines. The authors should use more efforts to describe this section along with suitable motivations.
Author Response
Reviewer 4
Comments:
The paper deals with Improving the Accuracy of Fault Frequencies by Means of LMDand Ratio Correction Methods for Rolling Bearing Failure. In spite of the incontestable appeal of the proposed approach, there are several points, which impair its quality:
1. First of all, the authors should avoid using uncommon shortcuts in the title.
Response: Many thanks for your careful check! We are sorry for our negligence. We have modified this mistakes in the revised manuscript.
2. Secondly, making an orthographic error in the title is not a good idea (Frequencies).
Response: Many thanks for your careful check! We are sorry for our negligence. We have modified this mistakes in the revised manuscript.
3. The fault frequencies are as they are and cannot be improved. One can only improve its estimation quality.
Response: Many thanks for your careful check!
4. The above comments should be also incorporated within the abstract.
Response: Many thanks for your comments! We have added this comments in the revised manuscript.
5. What is a particular meaning of “1” in eq. 1?
Response: Many thanks for your careful check! We are sorry for our negligence. We have added the explanation in the revised manuscript.
Spectrum correction is to find the abscissa of the center of the main lobe, as shown in Figure 1(a). Assume the spectrum of the window spectrum function is and its function expression is known. Besides, it is symmetrical about -axis. and are adjacent spectral lines and they are closest to the peak of frequency. Their corresponding window spectrum function are and . Their corresponding discrete spectrum are and . is the spectral line correction and it can be calculated by the following equation.
6. The authors should provide the existence condition of an inverse (3) (Improve equation quality as well).
Response: Many thanks for your careful check! We are sorry for our negligence. We have provided the existence condition of an inverse (4) in the revised manuscript.
where, the condition for the existence of the inverse function is that is in one-to-one correspondence with in equation (3). Then, the frequency correction value can be calculated according to . Therefore, it is called ratio correction method.
7. Delata x and Delat k denote the same thing. The authors should condense such a description.
Response: Many thanks for your careful check! We are sorry for our negligence. We have modified this mistakes in the revised manuscript.
8. What is the meaning of * in (5)?
Response: Many thanks for your careful check! We are sorry for our negligence. We have modified this mistakes in (6) in the revised manuscript.
9. The authors introduce modulation strategies without any prior background. The reader will be confused while analysing this part of the paper.
Response: Many thanks for your careful check! We are sorry for our negligence. We were aware of this problem and deleted this section in the revised manuscript.
10. The main contribution of the paper (Sec. 2.4) has just a few lines. The authors should use more efforts to describe this section along with suitable motivations.
Response: Many thanks for your comments! we added the more description about the main contribution of the paper(Sec. 2.3).
Round 2
Reviewer 2 Report
Significant improvements have been done by the authors. However, still there are some issues which have to be solved and improved. The comments are added below:
· The English academic writing has still some mistakes. I recommend English edition service. For instances:
- In line 41, keep the definition article, “the”. Any noun in English should carry a sign of definition.
- The expression you used in line 43-44, “play an important role at” does not have a proper preposition.
- Again line 95, the preposition “with” you used is incorrect.
- Using the “but” instead of “however” is not academically pleasant. So in line 308, change it to however.
- Apply the previous comment to the whole text.
· It seems that the citation style is different from one to another. You should stick to a same voice and avoid changing the style. Please be aware that this is an important comment. Also, Reference number 38 is incorrectly written. See the following link for the correct information: https://www.mdpi.com/2075-5309/8/9/128
· Delete the “not shown” statement you wrote in line 245. It does not make any sense.
· In line 287, I cannot understand what you wrote. What does “whether the fault frequency is corrected or not are not” mean?
Author Response
1. In line 41, keep the definition article, “the”. Any noun in English should carry a sign of definition.
Response: Many thanks for your careful check! We are sorry for our negligence. We keep the definition article, “the” in line 41, and we have modified similar mistakes about article in the revised manuscript.
Modified sentence: The third part is the noise and interference
2. The expression you used in line 43-44, “play an important role at” does not have a proper preposition.
Response: Many thanks for your careful check! We are sorry for our negligence. We have modified this mistakes in the revised manuscript, and we have modified similar mistakes about preposition in the revised manuscript.
Modified sentence: Although these methods are effective and useful in the fault diagnosis of rolling bearing, there are still some limits.
3. Again line 95, the preposition “with” you used is incorrect.
Response: Many thanks for your careful check! We are sorry for our negligence. We have modified this mistakes in the revised manuscript, and we have modified similar mistakes about preposition in the revised manuscript.
Modified sentence: Accurate frequency estimation is one of the most basic problems in the field of signal processing
4. Using the “but” instead of “however” is not academically pleasant. So in line 308, change it to however.
Response: Many thanks for your careful check! We are sorry for our negligence. We have modified this mistakes in the revised manuscript, and we have modified similar mistakes in the revised manuscript.
Modified sentence: However, it does not affect the accurate extraction of the fault frequency.
5. Apply the previous comment to the whole text
Response: Many thanks for your careful check! We are sorry for our negligence. We have modified similar mistakes in the revised manuscript.
6. It seems that the citation style is different from one to another. You should stick to a same voice and avoid changing the style. Please be aware that this is an important comment. Also, Reference number 38 is incorrectly written. See the following link for the correct information: https://www.mdpi.com/2075-5309/8/9/128
Response: Many thanks for your comments and careful check! We have made the citation style the same in the revised manuscript.
We have modified the mistake about reference number 38 in the revised manuscript.
Modified sentence: Aryan, R.R.; Mehdi, B. Probabilistic Risk-Based Performance Evaluation of Seismically Base-Isolated Steel Structures Subjected to Far-Field Earthquakes. Buildings 2018, 8(9), 128.
7. Delete the “not shown” statement you wrote in line 245. It does not make any sense.
Response: Many thanks for your careful check! We are sorry for our negligence. We deleted the “not shown” statement in line 245 in the revised manuscript.
8. In line 287, I cannot understand what you wrote. What does “whether the fault frequency is corrected or not are not” mean?
Response: Many thanks for your careful check! We are sorry for our negligence. We corrected this mistake in line 287 and other similar mistakes in the revised manuscript.
Modified sentence in line 287: In this case, whether it is corrected or not, the fault frequency is almost equal to the theoretical fault frequency. In other words, for the long-time signals, it does not affect the accurate extraction of fault frequency whether the fault frequency is corrected or not.
Modified sentence in line 347: In this regard, a more in-depth research is needed. However, when analyzing the long-time signals, it does not affect the accuracy of the fault frequency whether the fault frequency is corrected or not.
Modified sentence in line 397: For the long-time signals of rolling bearing, the fault frequency is almost equal to the theoretical fault frequency whether it is corrected or not.
Reviewer 4 Report
The authors improved the paper according to my guidelines and it can be accepted in its current form.
Author Response
Many thanks for your comments!